# Induction of SUSD2 by STAT3 Activation Is Associated with Tumor Recurrence in HER2-Positive Breast Cancer

**DOI:** 10.3390/cells14010019

**Published:** 2024-12-28

**Authors:** Yisun Jeong, Hyungjoo Kim, Daeun You, Soo Youn Cho, Sun Young Yoon, Seok Won Kim, Seok Jin Nam, Jeong Eon Lee, Sangmin Kim

**Affiliations:** 1Department of Breast Cancer Center, Samsung Medical Center, 81 Irwon-Ro, Gangnam-gu, Seoul 06351, Republic of Korea; sunrise1526@naver.com (Y.J.); hjook1990@gmail.com (H.K.);; 2Department of Surgery, Samsung Medical Center, Sungkyunkwan University School of Medicine, 81 Irwon-Ro, Gangnam-gu, Seoul 06351, Republic of Korea; 3Department of Pathology, Stanford University, Stanford, CA 94305, USA; 4Department of Pathology and Translational Genomics, Samsung Medical Center, Sungkyunkwan University School of Medicine, 81 Irwon-Ro, Gangnam-gu, Seoul 06351, Republic of Korea; 5Department of Health Sciences and Technology, SAIHST, Sungkyunkwan University, 81 Irwon-Ro, Gangnam-gu, Seoul 06351, Republic of Korea

**Keywords:** HER2, SUSD2, STAT3, trastuzumab resistance, EGFR+ HER2+

## Abstract

Sushi domain-containing protein 2 (SUSD2), a transmembrane protein containing a sushi motif, has been reported to have tumor-promoting functions in various types of cancer, including breast cancer. However, the regulatory mechanism of SUSD2 and its function in HER2-positive (HER2+) breast cancer have not been fully identified as yet. In this study, we explored the potential of targeting SUSD2 to overcome trastuzumab (TRZ) resistance in HER2+ breast cancer. SUSD2 expression was found to be significantly increased in HER2-overexpressing cells. Endogenous SUSD2 expression was observed in HER2+ breast cancer cells but not in estrogen receptor-positive or triple-negative breast cancer cells. We also found that SUSD2 expression was positively correlated with HER2 expression in a publicly available human primary breast cancer dataset. Although SUSD2 expression was associated with HER2, its expression levels were not affected by TRZ. Through kinase array experiments, we found that SUSD2 expression was modulated downstream of STAT3-dependent signaling in breast cancer cells overexpressing HER2. STAT3 activity was increased in EGFR+ HER2+ breast cancer cells compared to EGFR+ cells. Furthermore, we observed that SUSD2 expression was decreased by C188-9, a STAT3-specific inhibitor. Finally, we analyzed the association between patient survival and SUSD2 expression in breast cancer. Our results showed that SUSD2 expression had a negative correlation with the relapse-free survival of patients with EGFR+ HER2+ breast cancer when compared to EGFR+ breast cancer patients. Collectively, our results demonstrate that SUSD2 expression is mediated by STAT3 and imply the potential of using SUSD2 as a biomarker to stratify HER2+ breast cancer.

## 1. Introduction

Human epidermal growth factor receptor 2 (HER2) is overexpressed in 20–25% of all breast cancer cases [1]. Since HER2 is an oncogene with various roles in breast cancer progression, anti-HER2 agents have been used for the treatment of HER2-positive (HER2+) breast cancer [2]. Although HER2-targeted therapies, including antibody drugs and tyrosine kinase inhibitors, have been found to be effective through clinical trials, they possess limitations in that patients can show initial resistance or develop acquired resistance to those treatments [3]. Therefore, further exploring novel therapeutic targets for HER2+ breast cancer and delineating the mechanisms associated with resistance to anti-HER2 agents could improve the treatment of HER2+ breast cancer.

Previous studies have implied that the simultaneous expression of EGFR and HER2 is associated with breast cancer aggressiveness [4]. Cohen et al. have shown that a clone of NIH3T3 cells transfected to co-express EGFR and HER2 shows higher levels of tumor-forming abilities in nude mice than other conditions (EGFR, HER2, HER3, HER4, EGFR+ HER3+, EGFR+ HER4+, and HER2+ HER3+) [5]. Another study has assessed the survival rates of 670 breast cancer patients according to EGFR and HER2 expression levels and demonstrated that EGFR+ HER2+ patients have a worse disease-free survival and overall survival than EGFR- HER2-, EGFR+ HER2-, and EGFR- HER2+ patients [6]. Based on these previous results, we hypothesized that EGFR expression levels could affect the treatment of HER2+ breast cancer patients through dimerization with HER2+, further leading to differences in reactivity to anti-HER2 antibodies such as trastuzumab (TRZ) and trastuzumab emtansine (T-DM1).

Sushi domain-containing protein 2 (SUSD2) is a single-pass membrane protein that contains consensus sushi motifs in the C-terminus. It was originally identified as a gene expressed in normal rat cells [7]. SUSD2 expression is known to be altered in various types of cancers including colon cancer, non-small cell lung cancer (NSCLC), and gastric cancer [8,9,10]. Watson et al. have reported that SUSD2 is positively expressed in human breast cancer tissues and that it could increase breast cancer cell invasion [11]. Immunohistochemical analysis has revealed that SUSD2 levels are correlated with an increased presence of M2-polarized tumor-associated macrophages (TAMs), which have tumor-promoting functions in human breast tumor tissues [12]. SUSD2 can directly regulate the expression of chemokine (C-C motif) ligand 2 (CCL2), which is responsible for recruiting macrophages to the tumor microenvironment [12]. Breast tumor tissues with HER2 amplification show a higher staining intensity of SUSD2 than tissues of other breast cancer subtypes [12].

Although previous studies have demonstrated an association between SUSD2 and breast cancer progression, the specific role of SUSD2 in HER2+ breast cancer remains unclear. In this study, we explored the function of SUSD2 in the context of HER2+ breast cancer.

## 2. Materials and Methods

### 2.1. Cell Culture

MDA-MB231 human breast cancer cells, either transfected with an empty vector or engineered to overexpress human epidermal growth factor receptor 2 (HER2) using a retroviral pBMN vector, were kindly provided by Dr. Incheol Shin (Hanyang University, Seoul, Republic of Korea). MDA-MB231 cells with the empty vector served as controls. MDA-MB231 and JIMT1 cells were cultured in Dulbecco’s modified Eagle’s medium (DMEM; HyClone, Cytiva, Marlborough, MA, USA), while SKBR3 cells were grown in RPMI1640 medium (HyClone, Cytiva). The culture media were supplemented with 10% fetal bovine serum (FBS; HyClone, Logan, UT, USA), 2 mM L-glutamine, 100 IU/mL penicillin, and 100 μg/mL streptomycin. All cell lines were maintained in a humidified incubator set at 37 °C with 95% air and 5% CO_2_ [4].

### 2.2. Western Blots

Cells were lysed on ice using PRO-PREP™ Protein Extraction Solution (Intron Biotechnology, Inc., Gyeonggi-do, Republic of Korea) and centrifuged at 16,100× *g* for 15 min. The supernatant was collected, and equal amounts of protein were denatured by boiling for 5 min in Laemmli sample buffer. Proteins were separated on SDS–polyacrylamide gels and transferred onto polyvinylidene fluoride (PVDF) membranes (Millipore, Billerica, MA, USA). The membranes were blocked with 5% skim milk for 1 h at room temperature and incubated overnight at 4 °C with primary antibodies: anti-β-actin (Abfrontier, Seoul, Republic of Korea, LF-PA0207, 1:5000), anti-p-EGFR (Abcam, Cambridge, UK, ab40815, 1:5000), anti-total EGFR (Abcam, ab52894, 1:10,000), anti-p-HER2 (CST, 2241S, 1:1000), anti-total HER2 (Santa Cruz, TX, USA, sc33684, 1:5000), anti-SUSD2 (Abcam, ab182147, 1:10,000), anti-ER-α (Santa Cruz, sc8002, 1:1000), anti-p-STAT3 (Abcam, ab76315, 1:10,000), and anti-total STAT3 (Santa Cruz, sc-8019, 1:1000). Following washes, the membranes were incubated for 1 h at room temperature with horseradish peroxidase (HRP)-conjugated secondary antibodies (Cell Signaling Technology, Danvers, MA, USA). Protein bands were detected using ECL™ Prime reagent (Cytiva, Marlborough, MA, USA).

### 2.3. Immunofluorescence Microscopy

Cells were plated on slides and allowed to adhere overnight. Fixation was performed using 4% paraformaldehyde for 15 min at room temperature, followed by permeabilization with 0.1% Tween 20 for 2 min. After fixation and permeabilization, the cells were blocked with 10% BSA for 30 min at room temperature. Samples were then incubated with anti-HER2 antibody (Santa Cruz Biotechnology, sc33684, 1:50) overnight at 4 °C. This was followed by incubation with Alexa Fluor 488-conjugated secondary antibody (Invitrogen, 1:250) for 1 h at room temperature. Nuclei were counterstained with Vectashield H-1200 containing DAPI (Vector Laboratories, Burlingame, CA, USA). Phosphate-buffered saline (PBS) washes were performed between each step. Fluorescent images were acquired using an LSM780 confocal laser scanning microscope (Carl Zeiss, Jena, Germany).

### 2.4. Colony Formation Assays

For colony formation assays, SKBR3 and JIMT1 breast cancer cells were seeded into 6-well plates at a density of 2 × 10^3^ cells per well and incubated overnight at 37 °C. The following day, the cells were treated with 50 μg/mL trastuzumab (TRZ) or 10 μM C188-9 (Selleck Chemicals, Houston, TX, USA) and cultured for an additional 12 days. Colonies were fixed with 100% methanol, stained with 0.01% crystal violet, and visualized using a CK40 inverted microscope (Olympus, Tokyo, Japan) [13]. The quantification of clonogenic assays was counted using Image J 1.54d software.

### 2.5. Cell Cycle Analysis

Cell cycle analysis was conducted using flow cytometry with propidium iodide (PI) staining. Cells were collected and fixed in 70% ethanol for 20 min at room temperature. Following fixation, the cells were treated with 100 µg/mL DNase-free RNase A (Thermo Fisher Scientific, Waltham, MA, USA) at 37 °C for 30 min. After rinsing with phosphate-buffered saline (PBS), the cells were resuspended in fluorescence-activated cell sorting (FACS) buffer containing 50 µg/mL PI (Sigma-Aldrich, St. Louis, MO, USA). Analysis was performed on a BD FACS Verse flow cytometer (BD Biosciences, San Diego, CA, USA) [4].

### 2.6. Quantitative Reverse Transcription PCR (RT-qPCR)

Total RNA was extracted from cells using TRIzol reagent (Thermo Fisher Scientific, Waltham, MA, USA). Complementary DNA (cDNA) was synthesized from the isolated RNA using a cDNA Synthesis kit (Thermo Fisher Scientific) following the manufacturer’s protocol. Quantitative reverse transcription PCR (RT-qPCR) was performed with SYBR Green master mix (Bioline Ltd., London, UK) on an ABI 7900HT real-time PCR system (Foster City, CA, USA). Beta-actin (ACTB) served as the reference gene for normalization. Relative gene expression levels were calculated using the comparative CT method (2^−ΔΔCT^). Thermal cycling conditions were as follows: 50 °C for 2 min, 95 °C for 10 min, followed by 40 cycles of 95 °C for 15 s, 60 °C for 15 s, and 72 °C for 15 s. The primer sequences used were as follows: human ACTB forward (F), TCA CCA TTG GCA ATG AGC GGT T; reverse (R), AGT TTC GTG GAT GCC ACA GGA CT; human EGFR F, CAT GTC GAT CTT CCA GA; R, GGG ACA GCT TGG ATC ACA CT; human HER2 F, CAC TTC AAC CAC AGT GGC AT; R, ATT CAC ATA CTC CCT GGG GA; human IL1B F, GCC CTA AAC AGA TGA AGT GCT C; R, GAA CCA GCA TCT TCC TCA G; human IL6 F, AAT TCG GTA CAT CCT CGA CGG; R, GGT TGT TTT CTG CCA GTG CC; human IL8 F, AGG GTT GCC AGA TGC AAT AC; R, AAA CCA AGG CAC AGT GGA AC; human MMP-1 F, ATT CTA CTG ATA TCG GGG CTT TGA; R, ATG TCC TTG GGG TAT CCG TGT AG; human SUSD2 F, GTG CAC TTC GGA TCA TCG AC; R, CAG CCG TAG TCC GTG AAG AA [4].

### 2.7. Human Phospho-Kinase Antibody Array

Cells were seeded at a density of 1 × 10^6^ cells per plate in two separate 100 mm dishes. Whole-cell lysates were prepared using the appropriate lysis buffers, and 300 µg of protein from each sample was applied to the Proteome Profiler Human Phospho-Kinase Antibody Array (R&D Systems, Minneapolis, MN, USA). Following the manufacturer’s protocol, membranes were blocked and incubated with cell lysates mixed with a biotinylated detection antibody cocktail at 4 °C for 12–16 h. After washing with 1× washing buffer, the membranes were treated with streptavidin–HRP for 30 min. Protein phosphorylation was detected by exposing the arrays to X-ray film using enhanced chemiluminescence (ECL). Phospho-kinase levels were quantified by comparing signals to the positive control spots on the same membrane [14].

### 2.8. Clinicopathological Characteristics of Breast Cancer Patients

A total of 21 breast cancer patients with EGFR+ and/or HER2+ expression were included in this study. Patients’ clinicopathological factors, immunohistochemistry, biologic factors, treatment modalities, and recurrence times were obtained from the the Breast Cancer Center’s clinical database at Samsung Medical Center.

Tumor stage was determined based on the American Joint Committee on Cancer 6th Staging System. Estrogen receptor and progesterone receptor staining data were extracted from pathology reports. Staining was scored using the Allred score (AS), a method that could semi-quantitate the proportion of positive cells (scored on a 0–5 scale) and staining intensity (scored on a 0–3 scale), with a maximum score of 8. An AS > 2 was considered positive [4,15]. Clinicopathological assessments were performed for subtype classification by an experienced pathologist at the Department of Pathology, Samsung Medical Center. The use of human tissues in this study was approved by the Institutional Review Board of Samsung Medical Center, Korea (IRB number 2017-11-124), and informed consent was obtained for all tissue donors.

### 2.9. Tissue Specimens

In this study, we employed breast cancer tissue specimens from the Biospecimen Bank, which stores tissues collected from breast cancer patients who underwent surgical resection at Samsung Medical Center. Total RNA was extracted from tissue specimens using Trizol reagent, and cDNA was synthesized using a cDNA Synthesis kit. qRT-PCR was performed as previously described.

### 2.10. Immunohistochemistry (IHC)

Paraffin-embedded tumor tissues were collected from the Department of Pathology. EGFR immunostaining was considered positive if at least 10% of tumor cells exhibited moderate to strong membrane staining. HER2 positivity was indicated by a 3+ intensity score via IHC or by a gene amplification ratio ≥ 2.0 via FISH when the IHC showed a 1+ or 2+ result. Tumor samples with no staining, or less than 10% of tumor cells showing membrane staining were assigned a score of ‘0’ [4]. The tissue sections were deparaffinized using xylene, hydrated with graded dilutions of alcohol, and then immersed in 3% hydrogen peroxide solution to neutralize endogenous peroxidase activity. Next, a microwave was used to heat sections in a citrate buffer for antigen retrieval. Slides were incubated with monoclonal antibodies against EGFR (1:30, M7239, Novocastra, Newcastle upon Tyne, UK), HER2 (1:250, A0485, DAKO, Santa Clara, CA, USA), and SUSD2 (1:500, ab121214, Abcam) for 1 h at RT. After washing, tissue sections were incubated with biotinylated anti-mouse secondary antibody, followed by incubation with streptavidin–horseradish peroxidase complex [15]. Slides were washed and immunoreactions were detected using an Ultraview Universal DAB detection kit (Ventana Medical Systems Inc., Oro Valley, AZ, USA) and 3,3”-diaminobenzidine, followed by counterstaining with hematoxylin and bluing reagent [16]. A pathologist with expertise in immunohistochemistry (Soo Youn Cho, Samsung Medical Center, Seoul, Republic of Korea) performed the assessments. Images were digitally acquired using a Scanscope AT2 apparatus (Aperio Technologies, Vista, CA, USA) at 200× magnification [4].

### 2.11. SUSD2 Knockdown

Scrambled siRNA and SUSD2-specific siRNA were procured from Bioneer (Daejeon, Republic of Korea). MDA-MB231 cells overexpressing HER2 were plated in 6-well plates. Transfections were performed using Lipofectamine 2000 (Invitrogen, Carlsbad, CA, USA) following the manufacturer’s instructions. After transfection, cells were incubated in serum- and antibiotic-free medium for 24 h, followed by an additional 48 h incubation in fresh medium containing 10% FBS.

### 2.12. IL-1B ELISA

IL1B protein levels in conditioned media from cells transfected with scrambled or SUSD2-specific siRNAs were measured using an ELISA kit (R&D Systems, Minneapolis, MN, USA) following the manufacturer’s instructions. The secreted IL1B levels were quantified to assess the impact of SUSD2 knockdown.

### 2.13. Survival Analysis

Relapse-free survival was analyzed in breast cancer patients using the Kaplan–Meier plotter database [17], based on SUSD2 mRNA expression levels. The analysis utilized the ‘227480_at’ probe set and employed the ‘Auto select best cutoff’ option to categorize patients by SUSD2 expression. Patient stratification by breast cancer subtypes (luminal A, HER2+, and basal) was performed using the ‘Subtype—PAM50’ setting. To analyze our clinical data, the recurrence time of all 21 breast cancer patients with EGFR+ and/or HER2+ expression was obtained from the clinical database of the Breast Cancer Center at Samsung Medical Center. RFS was analyzed using Prism 8. Hazard ratios (HRs) with 95% confidence intervals and log-rank *p*-values were calculated and reported. Statistical significance was defined as *p* < 0.05.

### 2.14. Public Data Analysis

The public dataset (GSE1456, GSE19615) was obtained from the PrognoScan database [18]. Differential expressions of SUSD2 were compared between various breast cancer subtypes. The correlation between HER2 and SUSD2 was analyzed based on expression values in 115 patients. The distant metastasis-free survival (DMFS) was analyzed in all breast cancer subtypes or HER2+ breast cancer patients using the same dataset. We divided the patients into two groups according to SUSD2 expression level in the entire patient cohort or HER2+ patients and plotted Kaplan-Meier survival curves.

### 2.15. Statistical Analysis

Graphs were created using Microsoft Excel 2016 (Microsoft, Redmond, WA, USA) and GraphPad Prism 8 (GraphPad Software, La Jolla, CA, USA). Statistical analyses were performed using Student’s *t*-test (unpaired, two-tailed) and one-way ANOVA. All experiments were conducted independently at least three times. Data are expressed as the mean ± standard error of the mean (SEM), and differences were considered statistically significant at *p* < 0.05.

## 3. Results

### 3.1. SUSD2 Is Increased by HER2 Overexpression

To explore the effect of EGFR and/or HER2 expression in breast cancer, we performed a cDNA microarray. First, we confirmed the overexpression of HER2 in MDA-MB231 cells stably transfected with a HER2-overexpressing vector (HER2). We found that HER2 protein and mRNA levels were specifically upregulated in HER2 cells compared to empty vector-transfected (Vec) cells (Figure 1A–C). Through cDNA microarray analysis, we found that SUSD2 was significantly increased in HER2 cells compared to Vec cells (Figure 1D, left panel). The expression of not only SUSD2 but also various other oncogenes such as MMP-1, IL-8, IL-6, and IL-1β were also upregulated in HER2 cells, consistent with the microarray results (Figure 1D, right panel, and Figure 2A). Since we found that SUSD2 expression was increased in HER2-overexpressing cells, we examined SUSD2 expression levels in breast cancer cells with various subtypes. We found that SUSD2 protein expression was upregulated in SKBR3 and JIMT1, cell lines with intrinsic HER2 overexpression (Figure 2B). Conversely, SUSD2 protein expression was not observed in breast cancer cells with estrogen receptor-positive (ER+) or triple-negative breast cancer (TNBC) subtypes (Figure 2B). SUSD2 mRNA levels were also increased in the SKBR3 and JIMT1 cell lines, consistent with their protein levels (Figure 2C). Furthermore, in a publicly available breast cancer patient dataset (GSE1456), SUSD2 expression levels were found to be upregulated in HER2+ breast cancer subtypes, including the luminal B and HER2 subtypes (Figure 2D).

### 3.2. SUSD2 Expression Is Associated with Poor Prognosis of HER2+ Breast Cancer Patients

Next, we sought to investigate the association between SUSD2 expression and survival in various breast cancer subtypes. Using a patient survival database, we found that SUSD2 expression was inversely correlated with the survival of HER2+ breast cancer patients (Figure 3A). However, we could not observe any significant association between SUSD2 expression and survival in breast cancer patients with other subtypes such as the basal subtype (Figure 3B) or the luminal A subtype (Figure 3C). Furthermore, we observed a positive correlation between SUSD2 and HER2 expression in the GSE19615 dataset (Figure 3D). Based on the above findings, we speculate that increased levels of SUSD2 are associated with the poor prognosis of HER2+ breast cancer patients.

### 3.3. SUSD2 Expression Is Modulated by STAT3 Activity

Our previous results indicated that SUSD2 was upregulated in HER2-overexpressing cells and that such an upregulation was associated with the poor survival of HER2+ breast cancer patients. We therefore attempted to further delineate the association between HER2 and SUSD2. However, when TRZ was used to block HER2 signaling, we could not find alterations in SUSD2 mRNA or protein level (Figure 4A,B). TRZ treatment led to reduced cell growth and cell cycle progression in SKBR3 cells, which are sensitive to TRZ, while it had no effect on cell growth or cell cycle progression in JIMT1 cells (Figure 4C,D). Since TRZ known to primarily block HER2 homodimer signaling had no effect on SUSD2 expression and that SUSD2 levels were increased in cells expressing both EGFR and HER2 (Figure 1A,D and Figure 2A–C), these findings suggest that SUSD2 might be regulated by EGFR and HER2 heterodimer signaling rather than HER2 signaling alone.

To identify the regulatory mechanism responsible for upregulated levels of SUSD2 in HER2-overexpressing cells, the phosphorylation levels of various signaling molecules were analyzed using a human phospho-kinase array. We found that STAT3 phosphorylation was increased in HER2 cells (Figure 5A). STAT3, a transcription factor, is known to regulate the transcription of various genes, leading to cancer progression [19]. When C188-9, a specific STAT3 inhibitor, was treated into HER2 cells, we found that SUSD2 levels and cell cycle progression were suppressed (Figure 5B–D). Also, STAT3 inhibition suppressed SUSD2 levels as well as cell growth and cell cycle progression in SKBR3 and JIMT1 cells (Figure 5E–H). Collectively, these results indicate that SUSD2 is regulated by STAT3 activity downstream of EGFR and HER2 signaling.

### 3.4. IL-1B Expression Is Regulated by SUSD2 Expression Levels

Next, we analyzed phenotypic changes in HER2-overexpressing cells following SUSD2 knockdown. As shown in Figure 6A, SUSD2-specific siRNA significantly decreased SUSD2 mRNA levels. Notably, IL-1B mRNA expression, which was upregulated by HER2 overexpression, was also reduced by SUSD2 knockdown (Figure 6A). Furthermore, IL-1B protein expression was decreased in the conditioned media of SUSD2 knockdown cells (Figure 6B). We also investigated the effect of SUSD2 knockdown on the cell cycle and observed the induction of G0/G1 phase arrest (Figure 6C). These findings suggest that SUSD2 expression can influence both inflammatory response and cell cycle regulation in HER2+ breast cancer cells.

### 3.5. SUSD2 Expression Is Increased in EGFR+ HER2+ Breast Cancer Patients

Since our previous results demonstrated the upregulation of SUSD2 in EGFR+ HER2+ breast cancer cells (Figure 1D and Figure 2A–C), we aimed to further explore this association in clinical samples. Patients with EGFR+ HER2+ breast cancer exhibited significantly lower survival rates than those with EGFR+ breast cancer (Figure 7A and Table 1). Additionally, SUSD2 mRNA levels were higher in EGFR+ HER2+ breast cancer than in EGFR+ breast cancer (Figure 7B). IHC staining revealed elevated SUSD2 protein levels in tissues from a patient (Case 1) with EGFR+ HER2+ breast cancer compared to tissues from a patient (Case 2) with EGFR+ breast cancer (Figure 7C). To address the limitation of small sample sizes, we extended our analysis using the PrognoScan database. Patients with a high SUSD2 expression had poorer distant metastasis-free survival rates than those with a low SUSD2 expression (Figure 7D, left panel). Notably, a clearer separation in survival rate was observed when the analysis was restricted to HER2+ patients (Figure 7D, right panel). Although the sample size is limited, these clinical sample results further corroborate our findings on STAT3-mediated SUSD2 upregulation in EGFR+ HER2+ breast cancer, underscoring the clinical significance of SUSD2 expression.

## 4. Discussion

Breast cancer is commonly classified into molecular subtypes based on the expression status of molecular markers, including ER, progesterone receptor (PR), hormone receptors (HRs), and HER2 (Luminal A, HR+ HER2-; Luminal B, HR+ HER2+; HER2: HR- HER2+; TNBC: HR- HER2-) [20]. Among molecular subtypes, the HER2+ and TNBC subtypes are known to be associated with poorer patient survival than other subtypes [21]. HER2+ breast cancer also has a higher rate of brain metastasis [22]. Previous studies have indicated that the survival of HER2+ breast cancer patients could be influenced by the expression of other oncogenes, including EGFR [6,23]. Recently, we have reported that the survival rate of EGFR+ and HER2+ breast cancer patients is significantly lower than that of EGFR+ breast cancer patients [4]. In that context, we have demonstrated that the expression of CCL2, a cytokine that is responsible for tumor-associated macrophage recruitment, is induced by HER2 overexpression in EGFR+ breast cancer cells [4]. These studies imply that EGFR and HER2 heterodimer-dependent signaling could contribute to increased tumor aggressiveness. Furthermore, a previous study has shown that the overexpression of SUSD2 can increase the expression of CCL2 in MDA-MB231 cells (EGFR+) and that the knockdown of SUSD2 downregulates CCL2 levels in SKBR3 (EGFR+ HER2+) cells, implying that the SUSD2/CCL2 axis could lead to macrophage recruitment in the tumor microenvironment (TME) [12]. Interestingly, we found that CCL2 and SUSD2 were not downregulated by TRZ, implying that both CCL2 and SUSD2 were regulated downstream of EGFR and HER2 heterodimer signaling, with possible associations with TRZ resistance.

Previously, various reports have demonstrated the function of SUSD2 in cancer. Watson et al. have reported that SUSD2 is highly expressed in breast lesions and cancers and that such a high expression increases cell invasion [11]. The knockdown of SUSD2 in SMMC-7721 human HCC cells can increase growth and decrease apoptosis, whereas the overexpression of SUSD2 decreases tumor growth and increases apoptosis in HepG2 human HCC cells [9]. Hultgren et al. have demonstrated that breast tumors with increased levels of SUSD2 have higher levels of M2-polarized TAMs, which have pro-tumorigenic functions [12]. On the other hand, SUSD2 can interact with colon-derived SUSD2 binding factor, attenuating the growth of the colon cancer cells through decreasing cyclin D and cyclin-dependent kinase 6 [10]. Furthermore, the knockdown of SUSD2 can increase the migration of HGSOC cells [24]. Similarly, the basal level of SUSD2 is decreased in NSCLC tissues and inversely correlated with clinical stage and lymph node metastasis in NSCLC [8]. Decreased SUSD2 levels are associated with higher histological grades, clinical stages, and poorer patient survival in HCC and high-grade serous ovarian carcinoma (HGSOC) [9,24]. Zhang et al. have shown that the silencing of SUSD2 in human endometrial adenocarcinoma cells can induce senescence and apoptosis [25]. In gastric cancer, SUSD2 is one of the 21 genes overexpressed in patients with hepatic recurrence [26]. The knockdown of SUSD2 in gastric cancer cell lines can attenuate cell growth, invasion, and migration rates [26]. For the cancer-extrinsic roles of SUSD2 in the TME, it has been shown that SUSD2 can interact with interleukin-2 receptor (IL-2R) alpha and lead to the impairment of anti-tumor CD8+ T cell function [27]. These previous reports collectively demonstrate that SUSD2 serves two-sided functions, such as pro-tumoral or anti-tumoral ones, depending on cancer type, including hepatocellular carcinoma, breast cancer, renal cell carcinoma, lung cancer, and ovarian cancer [11,28,29,30,31]. In this study, we identified the role of SUSD2 in the prognostic aspect and verified how SUSD2 is regulated in EGFR+ and/or HER2+ breast cancer.

We found that SUSD2 levels were increased in an EGFR+ cell line overexpressing HER2 and that SUSD2 levels were associated with poor patient survival in HER2+ breast cancer. Additionally, SUSD2 levels were upregulated in SKBR3 and JIMT1 cell lines, which are EGFR+ HER2+ breast cancer cell lines. Through human kinase arrays, we found that STAT3 signaling was activated in EGFR+ HER2+ breast cancer and that the inhibition of STAT3 led to the downregulation of SUSD2 levels, indicating that STAT3 was responsible for upregulated SUSD2 levels in EGFR+ HER2+ breast cancer. Furthermore, STAT3 inhibitors such as C188-9 and Stattic not only reduced SUSD2 levels, but also suppressed cell growth and cell cycle progression in EGFR+ HER2+ breast cancer cell lines. Our results also demonstrated that SUSD2 knockdown significantly decreased IL-1B mRNA and protein levels, indicating a direct regulatory role of SUSD2 in IL-1B expression. This reduction in IL-1B, a pro-inflammatory cytokine involved in cancer progression, suggests that SUSD2 might influence inflammatory responses and tumor microenvironment dynamics in HER2+ breast cancer. Previous studies have shown a link between STAT3 and TRZ resistance. For instance, Sonnenblick et al. have demonstrated that the phosphorylation of STAT3 is associated with TRZ resistance, using protein and gene expression data from TRZ-treated patients [32]. Aghazadeh et al. have shown that the STAT3/HIF-1α pathway is responsible for attenuated PTEN levels that confer resistance to TRZ [33]. Wang et al. have reported that STAT3 activation is associated with resistance to TRZ emtansine (TDM-1), an antibody–drug conjugate consisting of TRZ and cytotoxic agent DM1 [34]. Our study provides new insights into the link between STAT3 signaling and resistance to TRZ. Although our study partially demonstrated the clinical implications of SUSD2 in EGFR+ HER2+ breast cancer by showing that SUSD2 conferred a poorer survival to EGFR+ HER2+ breast cancer patients and that it was associated with EGFR and HER2 signaling, further mechanistic validation is necessary. For example, chromatin immunoprecipitation assays could confirm the direct binding of STAT3 to the SUSD2 promoter, strengthening our understanding of this regulatory pathway. Additionally, in vivo studies using EGFR+ HER2+ breast cancer models will be crucial to explore both the cancer-intrinsic and -extrinsic functions of SUSD2.

## 5. Conclusions

Although currently available anti-HER2 therapies have greatly improved patient survival, they have limitations in treating HER2+ breast cancer in recurrent and/or metastatic settings. In this study, we found that SUSD2 was a novel target in EGFR+ HER2+ breast cancer. SUSD2 expression had negative effects on the prognosis of EGFR+ HER2+ breast cancer patients. We also demonstrated that SUSD2 was modulated by STAT3, downstream of EGFR and/or HER2 signaling. Furthermore, we have shown a positive correlation between HER2 and SUSD2 expression in clinical samples, supporting our in vitro results. Our findings suggest the clinical significance of SUSD2 as a biomarker for EGFR+ HER2+ breast cancer and highlight the use of STAT3-specific inhibitors for treating EGFR+ HER2+ breast cancer expressing SUSD2.

## Figures and Tables

**Figure 1 cells-14-00019-f001:**
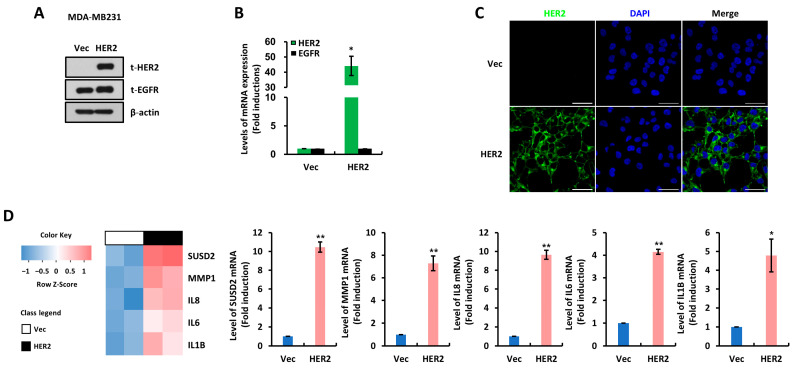
Sushi domain-containing protein 2 (SUSD2) is increased in HER2-overexpressing cells. (**A**) Western blots were performed using MDA-MB231 stable cell lines generated by transfection with empty vector (Vec) or HER2-overexpressing vector (HER2) with indicated antibodies. β-actin was used as loading control. (**B**) Quantitative reverse transcription PCR (RT-qPCR) was performed using MDA-MB231 Vec and HER2 cells to assess transcript levels of *EGFR* and *HER2*. Values were normalized to *ACTB*. Data represent at least three independent experiments. (**C**) Immunofluorescence microscopy was performed using MDA-MB231 Vec and HER2 cells. Scale bar: 100 μm. (**D**) Left, microarray was performed using MDA-MB231 Vec and HER2 cells. Right, RT-qPCR was performed using MDA-MB231 Vec and HER2 cells to assess transcript levels of indicated genes. Values were normalized to *ACTB*. Data are presented as mean ± SEM. *p*-values in (**B**,**D**) were calculated using Student’s *t*-test. * *p* <0.05 and ** *p* < 0.01.

**Figure 2 cells-14-00019-f002:**
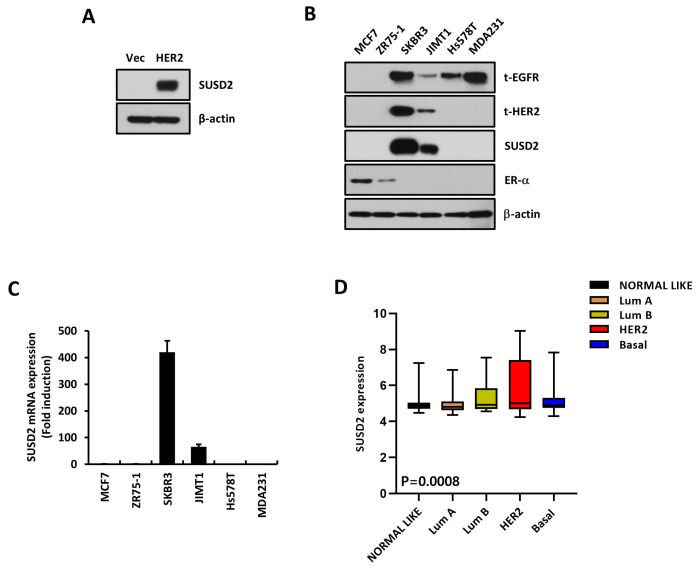
SUSD2 is overexpressed in HER2-positive (HER2+) breast cancer subtype. (**A**,**B**) Western blots were performed with MDA-MB231 Vec, HER2 cells (**A**), indicated cell lines (**B**), and indicated antibodies. β-actin was used as loading control. (**C**) RT-qPCR was performed with indicated cell lines to assess transcript levels of *SUSD2*. Values were normalized to *ACTB*. Data represent at least three independent experiments and are presented as mean ± SEM. (**D**) *SUSD2* levels were compared between breast cancer subtypes using public dataset (GSE1456).

**Figure 3 cells-14-00019-f003:**
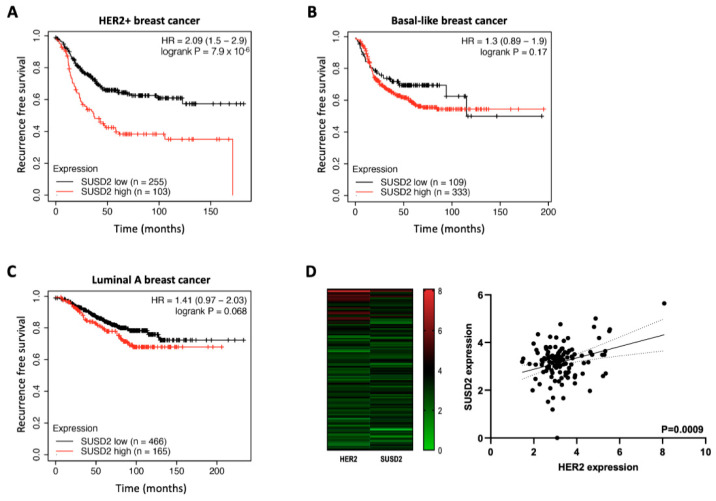
SUSD2 confers poor survival to breast cancer patients with the HER2+ subtype. Recurrence-free survival (RFS) was assessed using the Kaplan–Meier plotter database by stratifying breast cancer patients with the (**A**) HER2+ (low, *n* = 255; high, *n* = 103), (**B**) basal (low, *n* = 109; high, *n* = 333), and (**C**) luminal A (low, *n* = 466; high, *n* = 165) subtypes according to *SUSD2* expression. *p*-values were calculated using the log-rank test. (**D**) Left, heatmap showing HER2 and SUSD2 expression levels in breast cancer patients from the PrognoScan database (GSE19615, *n* = 115). Right, the correlation between HER2 and SUSD2 was analyzed from the same dataset.

**Figure 4 cells-14-00019-f004:**
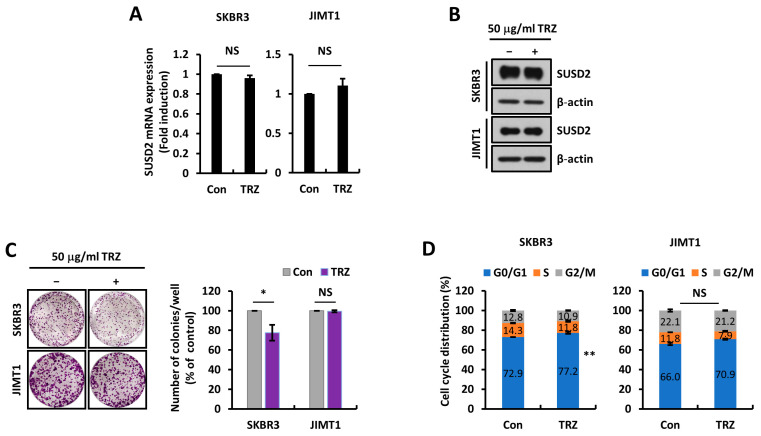
SUSD2 levels are not affected by trastuzumab (TRZ). (**A**) RT-qPCR was performed using SKBR3 and JIMT1 cells treated with control (Con) or TRZ to assess *SUSD2* transcript levels. (**B**) Western blots were performed using SKBR3 and JIMT1 cells treated with Con or 50 μg/mL TRZ. (**C**) Colony formation assays were performed using SKBR3 and JIMT1 cells treated with Con or 50 μg/mL TRZ (*n* = 3, left). Quantification of colonies number (right). (**D**) Cell cycle analyses were performed using SKBR3 and JIMT1 cells treated with Con or TRZ. NS: not significant, * *p* <0.05, and ** *p* < 0.01. Data represent at least three independent experiments and are presented as mean ± SEM.

**Figure 5 cells-14-00019-f005:**
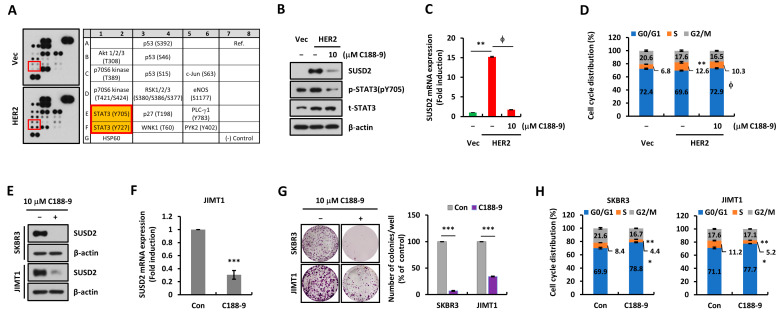
SUSD2 is regulated downstream of STAT3 activity. (**A**) A human phospho-kinase array was performed using MDA-MB231 Vec and HER2-overexpressing cells. (**B**) SUSD2 protein expression levels were analyzed by Western blots using MDA-MB231 Vec and HER2 cells treated with Con or 10 μM C188-9. β-actin was used as a loading control. (**C**) *SUSD2* mRNA expression levels were analyzed by RT-qPCR using MDA-MB231 Vec and HER2 cells treated with Con or 10 μM C188-9. Values were normalized to *ACTB*. (**D**) Cell cycle analyses were performed using MDA-MB231 Vec and HER2 cells treated with Con or 10 μM C188-9. (**E**) Western blots were performed using SKBR3 and JIMT1 cells after the treatment described in B. β-actin was used as a loading control. (**F**) RT-qPCR was performed using JIMT1 cells treated under identical conditions. Values were normalized to *ACTB*. (**G**) Colony formation assays were performed using SKBR3 and JIMT1 cells treated under identical conditions (*n* = 3, left). Quantification of colonies number (right), *** *p* < 0.001. (**H**) Cell cycle analyses were performed using SKBR3 and JIMT1 cells treated with Con or C188-9. * *p* < 0.05 and ** *p* < 0.01, Vec vs. vehicle-treated HER2; ϕ *p* < 0.05, vehicle-treated HER2 vs. C188-9-treated HER2 (in panel **C** and **D**). Data represent at least three independent experiments and are presented as the mean ± SEM.

**Figure 6 cells-14-00019-f006:**
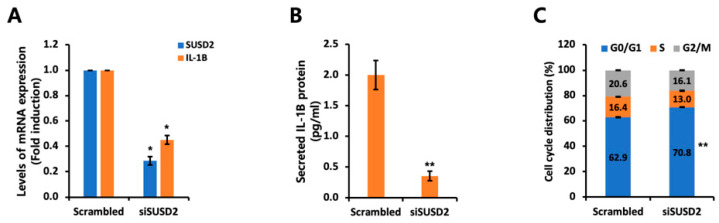
SUSD2 knockdown decreases IL-1B levels in MDA-MB231 HER2 cells. (**A**) Scrambled or SUSD2-specific siRNAs were transfected into MDA-MB231 HER2 cells. RT-qPCR was performed to assess *IL-1B* and *SUSD2* transcript levels. Values were normalized to *ACTB*. (**B**) Secreted IL-1B proteins were analyzed by ELISA in conditioned culture media of MDA-MB231 HER2 cells transfected with scrambled or SUSD2-specific siRNAs. (**C**) Cell cycle analyses were performed using MDA-MB231 HER2 cells transfected with scrambled or SUSD2-specific siRNAs. * *p* < 0.05 and ** *p* < 0.01. Data represent at least three independent experiments and are presented as mean ± SEM.

**Figure 7 cells-14-00019-f007:**
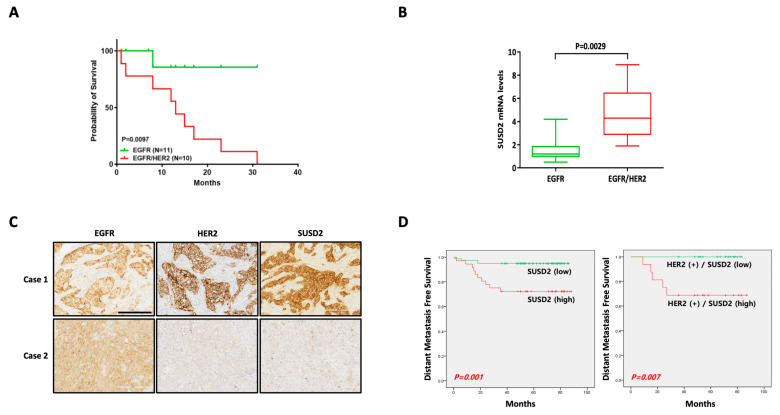
SUSD2 is upregulated in EGFR+ HER2+ breast cancer patients. (**A**) Recurrence-free survival (RFS) of patients with EGFR+ (*n* = 11) and EGFR+ HER2+ (*n* = 10) breast cancer. (**B**) *SUSD2* mRNA levels were compared between patients with EGFR+ breast cancer and those with EGFR+ HER2+ breast cancer (*n* = 21). (**C**) Immunohistochemistry was performed using tissues from EGFR+ and EGFR+ HER2+ breast cancer patients to assess expression levels of EGFR, HER2, and SUSD2. SUSD2 was expressed in both membrane and cytoplasm. Scale bar: 200 μm. (**D**) Distant metastasis-free survival rates were compared between breast cancer patients with high SUSD2 expression and those with low SUSD2 expression (*p* = 0.001) and between HER2+ breast cancer patients with high SUSD2 expression and those with low SUSD2 expression (*p* = 0.007). Data were extracted from PrognoScan database (left; *n* = 115, right; *n* = 36).

**Table 1 cells-14-00019-t001:** Clinicopathological characteristics of breast cancer patients.

Variables	All (*n* = 21)
Age group (years)	
>50	11 (52.4%)
≤50	10 (47.6%)
Subtypes	
LumA	6 (28.6%)
LumB	2 (9.5%)
HER2	8 (38.1%)
TNBC	5 (23.8%)
Estrogen receptor	
Negative	15 (71.4%)
Positive	6 (28.6%)
Progesterone receptor	
Negative	17 (81.0%)
Positive	4 (19.0%)
EGFR	
Negative	0 (0%)
Positive	21 (100%)
HER2	
Negative	11 (52.4%)
Positive	10 (47.6%)

## Data Availability

Data are contained within the article.

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
