# Peer review of "Induction of SUSD2 by STAT3 Activation Is Associated with Tumor Recurrence in HER2-Positive Breast Cancer"

_cells, 2024, doi:10.3390/cells14010019_

Round 1

Reviewer 1 Report

Comments and Suggestions for Authors

The manuscript by Jeong and colleagues investigated the relevance of Sushi domain-containing protein 2 (SUSD2) as a biomarker for types of breast cancer that express Human epidermal growth factor receptor 2 (HER2), as well as the importance of modulating the expression of this domain by STAT3, indicating the therapeutic potential of using inhibitors of this pathway for these cancer subtypes. The work was well conducted, demonstrating both experimental evidence in vitro and correlating with data obtained from patient samples. The experiments were rationally designed, supported by the use of various techniques such as microarray, RT-qPCR, as well as protein level verification using protein arrays and Western blotting. In this regard, the initially observed data were gradually confirmed, alongside counterproof using inhibitors. All original blotting images were provided in an attached file.

Minor: 

1- In lines 274-275, only one of the cell lines (JIMT1) is mentioned, but in figure 4 and its legend, both cell lines are described. 

2- In figure 5, item A shows a clear difference in the expression of p70S6 kinase (T389). Was this data investigated? Would it be interesting to discuss it?

Reviewer 2 Report

Comments and Suggestions for Authors

This article investigated the potential of targeting SUSD2 to overcome trastuzumab resistance in HER2+ breast cancer. The research team used many experimental approaches to address their question. The methodology is sound and well-described, and the results are written in an easy way that readers will be able to follow. Overall, this manuscript adds significant findings to the current literature. However, some comments need to be addressed in both method and results sections.

Methods:

1.        Immunohistochemistry (Line 169): please provide more information about the included patients in this analysis (the number of the patients, how did you select them, how did you categorize them into different molecular subtypes, did you do the ER, PR, Ki-67 to assist in the molecular classification). 

Please include which negative control you used for SUSD2 IHC staining and on which tissue on which it was applied.

2.        In the result section you mentioned some results taken from a public database (GSE19615). This database and its sources are not mentioned in the methods. Please add description of this database (accessed from which website, date of access, how many patients are included in this database, and of course a reference for this website).

Results

·       Figure 2 (line 246, 247): the terms (A, upper, B) and (A, lower, C) are unclear. Please clarify them specifying exactly which portions of the figure they refer to, as this phrasing is confusing.

·       Figure 2D: In this figure, it is clear that luminal B subtype of breast cancer demonstrate high expression of SUSD2. It would be valuable to mention this observation in the text, and one possible explanation that you can use that some of the luminal B subtype are triple positive (ER, PR, HER2 positive).

·       In Figure 3: you used KM plotter to analyze the overall survival (OS) of different molecular subtype based on SUSD2 expression. This is an excellent point, but I was expecting the Luminal B subtype to be included as mostly it will be also significant. So, it will a good addition to include luminal B to this section of the results.

·       Figure 5 (line 300-309): Please put the picture in order of how you will describe it as the way you used in this part is so confusing. e.g (B, F) just make it (B, C) and change the arrangements of the pictures.

Section 3.5 (Line 327): 

·      The sample size here is too small especially if we focus on the HER2 (only 8 patients). You can consider discussing this point in the limitation in the discussion.

·      Did you check the expression of HER2 and EGFR in all 21 patients or only in the HER2 positive patients. If you did the IHC analysis on all sample, please add comments on the expression of SUSD2 in all molecular subtypes.

·      Figure 7C: Clarify whether the three images in Figure 7C represent sections from the same patient, as the sections appear non-serial.

·      The pictures of IHC are not clear, please change the picture with high quality ones where the nuclear, cytoplasmic and membrane staining are clear.

Also, you need to include short comment on the IHC staining in the text that if the staining of SUSD2 is nuclear or cytoplasmic.

Reviewer 3 Report

Comments and Suggestions for Authors

In the manuscript titled “SUSD2 expression as a prognostic biomarker for HER2-positive breast cancer”, the authors have analyzed the functional role of SUSD2 in HER2-positive breast cancer cell lines as well as its prognostic role in a small number of clinical samples.

My major concerns are:

1.    The investigator has made an effort to examine the underlying mechanisms, yet the specifics of these mechanisms remain inadequately defined. A more comprehensive understanding of the precise interactions and pathways involved is needed to fully elucidate the role of SUSD2 in HER2+ breast cancer.

2.    The current title, "SUSD2 expression as a prognostic biomarker for HER2-positive breast cancer," may not fully capture the role of SUSD2 as a downstream target of STAT3 within the HER2 signaling pathway. Since SUSD2 expression is regulated by STAT3 activity, I suggest revising the title to reflect this relationship.

3.    My other primary concern is the declaration by the authors that SUSD2 is a prognostic marker in HER2-positive breast cancer. The sample size is too small to arrive at this conclusion. In addition, there appears to be no link between the functional studies conducted and the clinical analysis.

My other concerns are listed below.

1.    Introduction (Page 2, Line 78 – 83) – The authors have summarized their study findings. This is not needed in the Introduction. Instead, they should outline their study plan/aims.

2.    Materials and Methods  - There is no mention about the clinical samples with regards to where and how they were collected and if they were selectively chosen or randomly included, etc.

3.    Results (Section 3.5) – The authors mention “Since our previous results demonstrated upregulation of SUSD2 in EGFR+ HER2+ breast cancer”. Where are these previous results mentioned since I could not find it in the current manuscript. If it is from another previously published study, then reference needs to be cited here.

4.    Results (Section 3.5) – With regards to figure 7, the legends on the Y-axis are different for Fig 7A and Fig 7D. Were different survival parameters analyzed? Also, in the methods, the authors only mention relapse-free survival. However, in Fig 7D, they have mentioned distant recurrence-free survival. They need to correctly define what they mean by relapse/recurrence – is it overall relapse/recurrence or only distant relapse/recurrence? In addition, please include the number of cases in each group in Fig 7D (as done in Fig 7A).

5.    The current title, "SUSD2 expression as a prognostic biomarker for HER2-positive breast cancer," may not fully capture the role of SUSD2 as a downstream target of STAT3 within the HER2 signaling pathway. Since SUSD2 expression is regulated by STAT3 activity, I suggest revising the title to reflect this relationship.

6.    Please label the cell names clearly in Figures 1A, 2A, and 4D to provide clarity on the cell lines or types used in each figure.

7.    Fig 1C, Fig 7C. include and describe the scale bar in the legend.

8.    The SUD2 blot in Figure 2B appears overexposed, with significant background interference. Please replace it with a clearer blot that has reduced background noise to improve visibility and accuracy in assessing protein expression.

9.    Please provide the sample size (n) for each of the western blot experiments. Including the specific number of replicates or independent experiments (n) for each condition or treatment group will improve the transparency and reliability of the results.

10.   For Figures 4C and 5G, please quantify the results of the colony assay and specify the sample size (n) used for each condition.

11. Please incorporate a schematic diagram illustrating the signaling mechanism where EGFR and HER2 activate STAT3 phosphorylation, leading to the regulation of SUSD2.

Round 2

Reviewer 2 Report

Comments and Suggestions for Authors
